# Response of Plant Leaf Traits to Environmental Factors in Climax Communities at Varying Latitudes in Karst Regions

**DOI:** 10.3390/plants14020183

**Published:** 2025-01-10

**Authors:** Gang Xie, Yang Wang, Zhifeng Chen, Yangyang Ji, Yao Lu, Yi Liang, Ruiyu Zhou, Juan Tao

**Affiliations:** 1Institute of Guizhou Mountainous Resources, Guiyang 550001, China; 13628512665@163.com; 2College of Ecological Engineering, Guizhou University of Engineering Science, Bijie 551700, China; 19192682759@163.com (Z.C.); 15125497829@163.com (Y.J.); 16608803117@163.com (Y.L.); 13595773249@163.com (Y.L.); 18085091257@163.com (R.Z.); 18484643869@163.com (J.T.)

**Keywords:** latitudinal gradient, climax communities, functional traits, environmental factors, karst

## Abstract

Exploring the changes in plant functional traits and their relationship with the environment in karst climax communities across different latitudes can enhance our understanding of how these communities respond to environmental gradients. In this study, we focus on climax karst climax plant communities in Guizhou Province, China. We selected three sample sites located at varying latitudes and analyzed the variations in functional traits of the plant communities at these latitudes. Additionally, we examined the relationship between functional traits and environmental factors, integrating species characteristics and community structure into our analysis. The results indicated that (1) there were significant differences in both the community leaf aspect ratio and the community-specific leaf area. (2) Soil organic carbon content exhibited significant variations across different latitudes, while soil nitrogen content was notably higher in mid-latitude and low-latitude regions compared to high-latitude areas. The distribution of soil factors was more concentrated in high and mid-latitude regions, whereas low-latitude areas displayed more pronounced variability. (3) The primary environmental factors influencing the climax community in the karst study area included soil water content (SPMC), soil bulk density (BD), soil organic carbon content (SOC), soil nitrogen content (SNC), and soil phosphorus content (SPC). Our findings suggest that karst plant communities exhibit specific combinations of functional traits at distinct latitudes. With increasing latitude, the community demonstrated a gradual shift in ecological strategy from conservative to more opportunistic. Most environmental factors imposed limiting effects on plant functional traits, with plants primarily constrained by BD during growth. Among the responses of plant functional traits to environmental factors, community-weighted leaf area and community-weighted chlorophyll content were the most sensitive to soil conditions.

## 1. Introduction

Plant functional traits represent the expression of plant function and morphology under varying environmental conditions. The karst landscape is formed through the geological processes of water interacting with soluble rocks, such as carbonate rocks, gypsum, and rock salt. This interaction primarily involves chemical dissolution and erosion, complemented by mechanical actions, including water erosion, subduction, and collapse, as well as the phenomena resulting from these processes. Erosion is enhanced by mechanical processes such as flowing water, submerged erosion, and collapse, along with the phenomena resulting from these actions [1]. These traits reflect the ecological strategies of species and are frequently utilized to explain and predict community and ecosystem functions [2]. Leaves are essential functional organs in plant photosynthesis and are also the structures that interact most frequently with the surrounding environment. Investigating changes in leaf traits to understand plant adaptation to environmental conditions has become a significant method for studying the structure of plant communities [3]. The climax community refers to the community structure that emerges at the conclusion of vegetation restoration, and its characteristics significantly influence the maintenance and stability of forest ecosystems. Among the factors affecting plant functional traits, soil characteristics are a crucial component [4]. In studies of plant communities, both topographic and soil factors can determine the distribution of functional traits [5], and different soil characteristics can exert varying effects on these traits [6]. Within specific habitats, distinct ecological strategies may converge or diverge among different communities, and functional traits will respond accordingly to environmental conditions [7].

Karst areas are highly heterogeneous and characterized by fragility and complexity. The study of the genesis of plant communities in these regions is significant for understanding the adaptation of plant traits to habitat heterogeneity. Research has confirmed that the leaf functional traits of plants exhibit a pronounced gradient pattern along various environmental gradients, such as climate and soil. Additionally, there are notable differences in the functional traits of different plant species under identical environmental conditions [8]. As a gradient effect encompassing multiple environmental factors, examining the latitudinal patterns of species diversity is essential for exploring the dynamic responses of species to their environment, understanding the processes of community aggregation in different climatic zones, and predicting the future impacts of climate change on biodiversity [9]. Currently, most studies conducted by ecologists on the mechanisms of species coexistence in karst habitats are independent pieces of research that are scattered across various latitudinal belts.

The research conducted by Cheng et al. [10] on the genealogical and functional diversity of tropical cloud forest communities in Bawangling, Hainan Island, demonstrated that habitat filtering is a key factor driving community structure. Kemble et al. [11] found that the genealogical structure of Panama’s tropical rainforest plant communities gradually clusters with increasing spatial scales. Additionally, Li et al. [12], in their study of the species within evergreen-deciduous-broad-leaved mixed forests of the Central Subtropical Karst, discovered a strong correlation among different plant functional traits, various functional diversity indicators, and multiple functional redundancy indicators, which existed to varying degrees. In a study examining the phylogeny and functional trait structure of plant communities across different latitudes, Miao et al. [13] found that environmental filtering and competitive exclusion predominantly influenced species aggregation in most communities. Environmental factors such as soil water content, soil acidity and alkalinity, and average annual air temperature significantly impacted the pattern of plant traits. While the findings of these studies provide valuable insights into the community dynamics of their respective zones, the conclusions require further verification and expansion through subsequent research on communities at different latitudes, due to the lack of continuity in the scale of the studies.

This paper focuses on karst climax communities at various latitudes, examining how environmental changes along different latitudinal gradients contribute to regional variations in vegetation and environmental conditions. The study aims to investigate the following questions based on species, traits, and environmental factors. (1) How do the functional traits of karst climax communities vary along the latitudinal gradient? (2) What are the patterns of adaptation regarding functional traits and environmental factors in climax communities across different latitudinal gradients? By addressing these questions, we can evaluate the ecological and evolutionary processes that influence the ecological strategies of plants in karst top communities across varying latitudes.

## 2. Study Area and Research Methodology

### 2.1. Overview of the Study Area

Typical karst climax communities in Guizhou Province, China, were selected as the study sample (Figure 1). Field investigations revealed that the soil-forming rocks at the three sample sites were predominantly dolomite and limestone, with minimal soil coverage, more exposed rock surfaces, and shallow soil layers present only in depressions and valley bottoms. In the plant distribution area, the soil-forming rocks include sandstone, sandy shale, siliceous rock, and carbonate rock. The soil is primarily composed of siliceous-aluminous yellow–brown loam, followed by siliceous-aluminous yellow loam, yellow loam, and yellow–brown loam, with the soil layer at the top of the ridge measuring 20 to 30 cm in thickness. We established two criteria for selecting the apex community: first, our research team had previously studied these plots, and second, the biomass of the apex community was the most significant in the entire forest, exhibiting greater species richness, and the plants were larger and less individually differentiated. The external characteristics of the vegetation indicated that trees were the dominant species, while shrubs and herbaceous plants were relatively scarce. After consulting with the staff of the nature reserve, we identified the climax communities in different reserves and used them as sample sites for this study. The climate of the Dashahe area is characterized by high humidity and year-round fog and an annual average temperature of 17.7 °C, with annual precipitation ranging from 1200 to 1360 mm, high relative humidity, and a humid monsoon climate typical of the northern subtropics. The Yuntai Mountain area has an annual average temperature of 14.1 °C, with an extremely high temperature of 39.4 °C and a frost-free period of 209 days; the annual precipitation is 943.3 mm. The Maolan area experiences a middle subtropical monsoon temperate climate, with an average annual temperature of 15.3 °C, an average annual temperature variation of 18.3 °C, and a growing season lasting 237 days. The annual precipitation in this area is 1752.5 mm, and the average annual relative humidity is notably high.

The minimum sample area was determined to be 900 m^2^ (30 m × 30 m) using the ‘species-area curve’ method. To ensure ecological significance, three sample plots (replicates) were established for each climax stage, resulting in a total of nine sample plots covering an area of 8100 m^2^. Initially, we set up the sample plots and assessed the diversity of the plant communities in the study area. During the peak of the growing season, we collected plant leaves and soil samples from the designated plots.

### 2.2. Sample Setup

The community composition survey referenced the methods of Fang et al. [14]. In each sample plot, nine small sample squares were established, and each square was surveyed for trees, shrubs, and herbs. Specifically, nine tree samples were designated within each plot, with each tree sample covering an area of 10 m × 10 m. Four shrub samples were arranged in each plot along the diagonal, with each shrub sample occupying an area of 5 m × 5 m. Additionally, one herb sample was established within each shrub sample, covering an area of 2 m × 2 m. Thus, in one sample plot, we collected nine tree samples, each measuring 10 m × 10 m, four shrub samples measuring 5 m × 5 m, and four grass samples measuring 2 m × 2 m. We surveyed and recorded all species within the sample plots and measured their respective traits.

The basic information for each sample plot at the terminal stage is presented in Table 1.

### 2.3. Sample Collection and Processing

(1)Plant leaf sampling and processing

Leaf collection is divided into two parts. The first part involves collecting plant leaves from all species present at the sampling site for trait determination. The second part focuses on collecting leaves from the dominant species at the site for nutrient analysis. Prior to this, the dominant species (or those with significant importance values) in each stratum of the community were identified by integrating preliminary plant surveys with species diversity data analysis. Subsequently, samples were collected. In each 30 m × 30 m sample plot containing tree species in the arboreal layer, three to five healthy plants were selected as sample plants. Leaves from the four cardinal directions of the crowns of these sample plants were collected using high-pruning shears, resulting in a total of 27 arboreal sample leaves. The arborvitae leaves were divided into two portions and bagged. A portion of the samples was used to measure blade thickness using vernier calipers. To minimize measurement errors, leaf thickness was recorded on the same day the samples were collected. Additionally, 10 to 20 leaves were prepared for determining leaf area, leaf length, specific leaf area, and other relevant parameters. The second portion was placed in a sealed bag, transported to the laboratory, and dried in an oven at 75 °C until a constant weight was achieved. Subsequently, the dried leaves were crushed in a pulverizer, ground, sieved, labeled, and stored in bags for the analysis of soil organic carbon content (SOC), soil nitrogen content (SNC), and soil phosphorus content (SPC).

(2)Soil Collection

Selection of soils corresponding to the vegetation types of plant communities in the study area. A diagonal sampling method was employed to identify five sampling points within each sample plot, focusing on the soil layer of 0–20 cm. Due to the thin soil layer characteristic of karst areas, the actual depth may be less than 20 cm; thus, the specific depth during sampling shall prevail. The soil samples were mixed in equal volumes to create a composite sample, which was then placed into a sealed bag for elemental analyses, including soil organic carbon content (SOC), soil nitrogen content (SNC), and soil phosphorus content (SPC). Concurrently, a ring knife was used to collect samples for assessing soil water content and soil bulk density. The retrieved soil samples were transported to the laboratory, where gravel and debris were removed. The samples were then air-dried, ground, sieved, bagged, and labeled. The ring knife was dried at 100 °C until a constant weight was achieved, after which the dry weight was determined.

### 2.4. Determination of Functional Properties and Analysis of Samples

(1)Elemental Determination

The organic carbon content of both plant and soil samples was determined using the potassium dichromate oxidation-external heating method [15]. Plant samples were digested using the H_2_SO_4_-H_2_O_2_ method, while total nitrogen was measured by the indophenol blue colorimetric method (NY/T2017-2011) [15]. Total phosphorus was assessed using the molybdenum-antimony colorimetric method (NY/T2017-2011) [16]. For soil samples, total nitrogen was measured using the Kjeldahl method (LY/T1228-2015) [17], and total phosphorus was determined by the NaOH melting-molybdenum-antimony colorimetric method (LY/T1232-2015) [18]. Additionally, the potassium content in both plant and soil samples was analyzed using flame spectrophotometry.

(2)Determination of Functional Traits

Based on the principles of plant growth characteristics, resource acquisition, nutrient content and distribution, ease of trait determination, and reliability, six traits were selected for the assessment of functional traits (Table 2). The determination methods were referenced from the new manual for standardized measurement of functional traits in plants worldwide [19]. Leaf thickness was measured using electronic vernier calipers (Deli, DL91150, Shanghai, China), while chlorophyll content was assessed with a chlorophyll meter (caaKEr, MLCK-A, Beijing, China). Leaf fresh weight was recorded using a 1 in 10,000 electronic analytical balance, and leaf dry weight was determined after drying the samples in an oven and weighing them again with the same analytical balance (Leqi, YT1004, Shanghai, China). Leaf length, width, and area were scanned and calculated using a scanner in conjunction with Photoshop software 13.0 (HP, HPScanJet N92120, Shanghai, China). Additionally, the carbon (C), nitrogen (N), and phosphorus (P) contents, along with their ratios, were obtained through direct measurements.

### 2.5. Data Processing

The community-weighted mean (CWM) of functional traits in karst plant communities was derived from the weighted average of species’ functional trait values and their relative abundance [20]. The formula for its calculation is as follows:$$\mathrm{CWM}={\sum_{i=1}^{s}}_{P_{i}\times V_{i}}$$
where “S” represents the number of species in the community, “P_i_” denotes the relative abundance of species “i”, and “V_i_” indicates the value of a functional trait characteristic of species “i”. The importance value of the arbor layer was calculated as (relative abundance + relative frequency + relative dominance based on diameter at breast height)/3 and shrub and grass layers as (relative abundance + relative frequency + relative cover)/3.

The data were initially collated using Microsoft Excel 2019 and tested for normality and variance using the Kolmogorov–Smirnov normality test before analysis. Data were analyzed using SPSS version 25.0 [21], employing one-way ANOVA and Tukey’s HSD for multiple comparisons to assessing variations in leaf functional traits, soil characteristics, and other factors across latitudinal gradients in plant communities. Pearson correlation analysis was conducted to elucidate the relationships between these metrics, with the data presented as mean ± standard deviation. To further investigate the variation in plant community leaf functional traits and soil factors, we utilized the ‘ggcor’, ‘vegan’, and ‘ggplot2’ software packages in R version 4.3.2 for correlation analyses. We analyzed violin plots using the ‘ggpubr’ and ‘cowplot’ packages. Additionally, we conducted Redundancy Analysis (RDA) of environmental factors and plant leaf functional traits across latitudinal gradients using the Vegan software package. Furthermore, we employed ‘WGCNA’ and ‘igraph’ for network analyses and ‘reshape’ for bubble plot analyses. All statistical analyses and visualizations were performed using R version 4.3.2 [22].

## 3. Results and Analysis

### 3.1. Changes in Leaf Functional Traits of the Climax Community at Varying Latitudes

As shown in Figure 2, the chlorophyll content of the karst climax community exhibited significant variations across latitudes, with low latitude (36.42) > high latitude (34.97) > mid-latitude (30.29). Leaf thickness was significantly greater at high latitudes (0.25) compared to the other latitudes (both at 0.17). The community leaf area demonstrated a significant difference between mid-latitude and low latitude, with low latitude (22.07) > mid-latitude (20.15). The leaf aspect ratio was significantly different, showing mid-latitude (4.30) > low latitude (3.57) > high latitude (2.79). Significant differences were also observed in the specific leaf area, with low latitude (346.24) > mid-latitude (307.81) > high latitude (199.47). Community leaf tissue density was significantly different between mid-latitude and low latitude, with mid-latitude (2.65) > low latitude (1.61). The distribution of chlorophyll was more dispersed between low and mid-latitudes, while it was more concentrated at low latitudes. The distribution of leaf thickness was more concentrated in the high-latitude cluster and more dispersed in the mid- and low-latitudes. Chlorophyll distribution was more widely dispersed in the same regional assay, showing distinct dispersion values in high and mid-latitude communities, while it was more dispersed in low-latitude areas. In contrast, the distribution of leaf aspect ratios was more concentrated, displaying distinct discrete values for high- and low-latitude communities, whereas it was more dispersed in mid-latitude regions.

The distribution of specific leaf area was more concentrated, exhibiting distinct discrete values for the high-latitude and mid-latitude clusters, while it was more dispersed in the low-latitude region.

### 3.2. Correlations Between Leaf Traits in the Climax Community

Leaf area exhibited a highly significant positive correlation with specific leaf area and chlorophyll content (Figure 3). In contrast, specific leaf areas demonstrated a highly significant negative correlation with leaf tissue density, chlorophyll, leaf thickness, and leaf aspect ratio. Additionally, leaf tissue density showed a highly significant negative correlation with leaf thickness. Chlorophyll content was found to have a highly significant positive correlation with leaf thickness. Finally, leaf thickness exhibited a significant positive correlation with leaf aspect ratio.

### 3.3. Changes in Environmental Factors of the Climax Community at Different Latitudes

As illustrated in Figure 4, the soil capacity of the karst climax community exhibited significant variations across different latitudes, with mid-latitude (1.30) > high latitude (1.17) > low latitude (0.68). Soil water content also demonstrated notable differences among latitudes, as indicated by low latitude (31.72) > high latitude (17.38) > mid-latitude (15.75). Furthermore, soil organic carbon content varied significantly across latitudes, with low latitude (22.37) > mid-latitude (11.28) > high latitude (4.31). Soil nitrogen content was significantly higher in mid-latitude (6.66) and low latitude (6.20) compared to high latitude (2.50). Soil phosphorus content showed significant differences across latitudes, with low latitude (1.60) > high latitude (0.74) > mid-latitude (0.57). The SCN was significantly higher at low latitude (3.62) than at mid-latitude (1.69) and high latitude (1.71). The SCP was significantly greater in low and mid-latitude (19.73) than in high latitudes (5.73). Additionally, the SNP was significantly higher in mid-latitude (11.68) compared to high latitude (3.33) and low latitude (4.47). The distribution of soil factors was more concentrated in the high and mid-latitude regions, while the low-latitude region exhibited more significant discrete values.

### 3.4. Relationships Between Environmental Factors and Leaf Traits in the Climax Community

As demonstrated by the Redundancy Analysis (Figure 5), the first axis of environmental factors explained 77.27% of the variance in plant functional traits, while the first two axes of the Redundancy Analysis (RDA) accounted for a total of 88.75% of the explained variance. The primary environmental factors influencing the climax community in the karst study area included soil water content (SPMC), soil bulk density (BD), soil organic carbon content (SOC), soil nitrogen content (SNC), and soil phosphorus content (SPC). Their relative influence, in descending order, was as follows: BD > SPMC > SPC > SNC > SOC. From the distribution of the sample sites, it is evident that in the low-latitude region, the specific leaf area of the community exhibited a positive correlation with soil phosphorus content, soil water content, and soil organic carbon. Additionally, soil phosphorus content and soil organic carbon were positively correlated. The first two axes of the Redundancy Analysis (RDA) explained 88.75% of the total variance, indicating that soil phosphorus content and soil moisture content were the primary environmental factors influencing the area. In the mid-latitude region, community-weighted leaf area and leaf tissue density exhibited a positive correlation with soil bulk density, which was identified as the predominant environmental factor. In high-latitude areas, community leaf thickness, leaf aspect ratio, and leaf dry matter mass exhibited strong correlations and were more concentrated. Furthermore, chlorophyll emerged as a prominent community trait in high-latitude regions, displaying a negative correlation with soil nitrogen content.

The network analysis of correlations between community traits and leaf-soil nutrients is illustrated in Figure 6A. Measurements with a *p*-value of less than 0.05 are connected by correlation lines, with red lines indicating positive correlations and blue lines indicating negative correlations. These measurements are categorized into three groups: community traits, leaf nutrients, and soil nutrients. The segments of the correlation lines were counted, resulting in the creation of Figure 6B. As shown in Figure 6B, the size of the circular bubbles represents the number of correlation lines among the three categories, with specific values labeled in the figure. A total of 50 correlation lines were identified, of which 4 (8%) were between community traits and leaf nutrients. Additionally, there were 5 (10%) correlations between community traits and soil nutrients, 13 (26%) between leaf nutrients and soil nutrients, and 28 (56%) correlations among similar measurements (calculated as 4 + 7 + 17).

## 4. Discussion

### 4.1. Changes in the Laws of Plant Functional Traits in Karst Vertex Communities Across Different Latitudes

The results of this study indicated that the community-weighted mean specific leaf area decreased significantly with increasing latitude, which aligns with the latitudinal distribution patterns of leaf traits identified by Reich [23] and Wright [24]. A lower specific leaf area (SLA) suggests that the plant community is more adept at utilizing environmental resources [25]. The study area is characterized by a typical karst plateau rocky desertification landscape, which has relatively limited soil and water resources. As plant traits change, leaves enhance water use efficiency by minimizing water loss due to transpiration. Additionally, plant communities improve their adaptability to the environment by optimizing nutrient utilization and conserving water. These findings are consistent with the conclusions drawn by Liu et al. [26], Zhang et al. [27], and Li et al. [28]. As latitude increases, hydrothermal conditions diminish, leading to increased environmental stress.

High-latitude polar communities are predominantly characterized by coniferous forests, where coniferous leaves are adapted to minimize water loss by increasing leaf thickness. In this study, the community-weighted mean leaf thickness (CWM.LT) decreased with decreasing latitude, while the community-weighted mean leaf area (CWM.LA) increased with decreasing latitude (Figure 2C). This suggests that plants enhance their leaf surface area by reducing leaf thickness (LT) and increasing leaf area (LA) to effectively capture light resources [29]. Additionally, communities characterized by thinner leaves were observed at lower latitudes, a condition attributed to plant adaptations aimed at minimizing aerobic respiration and conserving nutrients. Furthermore, the results indicated that leaf tissue density was significantly higher in mid-latitudes compared to other regions. This phenomenon is likely due to the pronounced impact of human activities on areas outside the mid-latitude apex community [30]. Species within these stands adapted to their environment under stressful conditions primarily by developing mechanical resistance, reducing nutrient cycling, and employing other defensive strategies against high-input pressures. These strategies included decreasing leaf area (LA), enhancing photosynthetic rates, and increasing leaf thickness density (LTD) to cope with the resource-poor peak scrub landscape characteristic of the region [31]. Species within these stands have adapted to environmental stressors primarily by enhancing their mechanical resistance, reducing nutrient cycling, and employing various defense strategies against high-input pressures (e.g., decreasing leaf area, increasing photosynthetic rates, and enhancing leaf tissue density). These adaptations are essential for survival in the resource-limited peak scrub landscape of the region [32]. Our results indicate that CWM.CHL, CWM.LT, CWM.LA, and CWM.LTN exhibit insignificant changes in trait characteristics across latitudes, demonstrating trait convergence. Trait convergence typically arises from environmental filtering and competitive exclusion. Specifically, environmental pressures can limit the range of viable species, leading to similar traits among different species that adapt to the environment. This phenomenon occurs due to environmental filtering, resulting in trait convergence at the community level [33]. Additionally, trait convergence may also arise when competitively advantaged species with similar trait values exclude competitively disadvantaged species with differing trait values [34].

### 4.2. Changes in Soil Factors in Karst Vertex Communities at Varying Latitudes

Karst landscapes represent a fragile ecological environment characterized by high habitat heterogeneity and complex geomorphology. The distribution and changes in soil nutrients significantly influence plant functional traits, as well as their growth and development [35,36]. The nutrient content of soil fractions serves as a crucial indicator for characterizing the composition and quality of soil organic matter [37]. The SCN can be utilized to assess the rate of decomposition of soil organic matter, while the SCP indicates the availability of effective phosphorus in the soil. Additionally, the SNP is the most effective indicator for predicting nutrient limitations in forest ecosystems.

In this study, the SCN, SCP, and SNP of the climax community soils were found to be 2.33, 14.41, and 6.50, respectively. These values are relatively low compared to global forest soils [38] (14.5, 211, and 14.6) and national terrestrial surface soils [39] (14.4, 136, and 9.3). This finding aligns with the study by Liu et al. [40], which suggests that the soil phosphorus levels are high while nitrogen levels are deficient, indicating an N-limited type of soil. When comparing soils across different latitude climax communities, it was observed that low-latitude areas exhibit higher soil water content, soil organic carbon, and soil phosphorus content, with significantly higher C:N ratios than other regions. The harsh living environment of the Maolan Karst forests, which grow on dolomite and limestone, results in very little soil presence and a high ratio of bare rock. Soil is primarily found in depressions and at the bottoms of valleys, where a shallow layer of soil exists. Additionally, scattered humus soil can be found in some stone gullies and crevices [41]. The soil surface is often covered with dead leaves, contributing to a soft, moist, and organic matter-rich environment. The distribution of soil factors is more concentrated in high and mid-latitude areas, while low-latitude areas exhibit more pronounced discrete values [42]. 

### 4.3. Response of Plant Functional Traits in the Climax Community to Environmental Factors at Different Latitudes

The environmental factors that decisively influence the distribution of plant functional traits are usually different at different scales, and the distribution of functional traits at a given site is often the result of cascading filtration from large to small scales as well as the combined effects of multiple factors [43]. The distribution patterns of community functional traits along spatial environmental gradients at large scales often stem from differences in functional traits within communities [7,24].

In this study, Redundancy Analysis (RDA) demonstrated a strong correlation between soil factors and leaf functional traits, consistent with the findings of Wang et al. [44]. Key environmental factors influencing plant functional traits in the climax community of the karst study area included SNC, BD, SPC, SPMC, and SOC (Figure 5). In low-latitude areas, these environmental factors were influenced by a greater number of additional environmental variables, leading to a stronger dependence of plant nutrient content on soil nutrients (Figure 6). Some of these nutrients were utilized for plant growth and organic matter accumulation, while others helped plants resist nutrient deprivation. The impact of total soil nitrogen was particularly pronounced in low-latitude regions, where communities dominated by glossy balsam and balsam trees exhibited a conservative strategy characterized by a combination of low specific leaf area and leaf thickness [32]. The results of this study also indicated that the functional traits of plant communities at low latitudes may be influenced by soil water content, soil phosphorus content, and soil organic carbon, particularly affecting changes in community-specific leaf area. This finding aligns with the research conducted by Zhou et al. [45]. Specific leaf area serves as a functional trait indicator that characterizes the interactions between plants and their environments; it is generally smaller in barren or harsh conditions and larger in resource- and nutrient-rich environments. Furthermore, both intraspecific and interspecific specific leaf areas tend to increase with rising temperatures but decrease with increased solar radiation, indicating that plant leaves are thinner in warmer environments and thicker in well-lit conditions [46]. Our study revealed a significant decrease in specific leaf areas with increasing latitude, suggesting that habitat barrenness intensifies in karst climax communities as latitude increases. Under similar conditions, temperatures in the karst climax community decline with increasing latitude. The functional traits of plant communities in mid-latitudes may be influenced by soil nitrogen content and soil bulk density, particularly affecting community leaf area and leaf tissue density. The size of the leaf area directly impacts photosynthetic efficiency, which, to some extent, reflects the level of plant production [41]. Mid-latitude regions are more susceptible to human activities, and species within these stands primarily adapt to environmental stressors by developing mechanical resistance, reducing nutrient cycling, and employing other defensive strategies against high input pressures (e.g., reducing leaf area, enhancing photosynthesis rates, increasing leaf thickness density, etc.) [47].

The functional traits of plant communities in high-latitude regions may be influenced by soil nitrogen content and soil organic carbon levels. However, changes in the functional traits of these plant communities—excluding chlorophyll—are not significantly driven by soil factors. This suggests that the structure and complexity of plant communities result from a combination of multiple soil factors rather than the influence of a single soil factor [48].

In this study, the primary soil factors were ranked as follows: BD > SPMC > SPC > SNC > SOC. This ranking may be attributed to the shallow and thin soil layers characteristic of karst areas, which exhibit limited water and fertilizer retention capacity. Additionally, soil erosion can deplete both water and soil organic carbon, resulting in reduced water content and organic carbon levels in the study area. The explanation rate of environmental factors on functional traits, as illustrated in Figure 6, supports this observation. In karst regions, environmental heterogeneity significantly influences plant leaf traits, which can vary considerably both between and within species. Generally, these traits exhibit a combination that results in lower specific leaf area and leaf area. This trait combination suggests that plants are likely to develop a set of drought-resistant traits to adapt to the physiological drought induced by habitat characteristics, such as shallow karst soils and soil water leakage.

## 5. Conclusions

(1)Karst plant communities display distinct functional trait combinations at varying latitudes. As latitude increases, the resource utilization traits shift to a combination characterized by high leaf thickness (LT) and low specific leaf area (SLA). In terms of overall changes, the functional traits of the apex communities exhibited some convergence effects, which may be attributed to the limited variation in environmental pressures with latitude in the karst region;(2)Bulk density (BD), soil microbial carbon (SPMC), soil nitrogen carbon (SNC), soil phosphorus content (SPC), and soil organic carbon (SOC) play a significant role in the changes in functional traits during the process of plant restoration. Plants are primarily limited by bulk density (BD) during growth, and most environmental factors exert a limiting effect on plant functional traits. Furthermore, many plant functional traits can only fulfill their roles under specific environmental conditions. Among the various responses of plant functional traits to environmental factors, community-weighted mean leaf area (CWM.LA) and community-weighted mean chlorophyll content (CWM.CHL) were found to be the most sensitive to soil factors.

## Figures and Tables

**Figure 1 plants-14-00183-f001:**
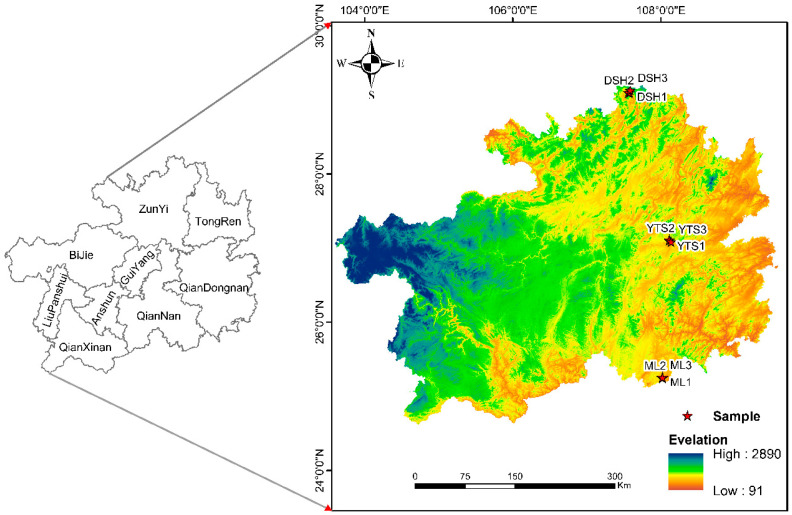
Schematic diagram of the distribution of sample sites in the study area. DSH represents the sample site in the high-latitude region of Dashahe, YTS reaches the sample site in the mid-latitude region of Yuntai Mountain, and ML represents the sample site in the low-latitude region of Maolan.

**Figure 2 plants-14-00183-f002:**
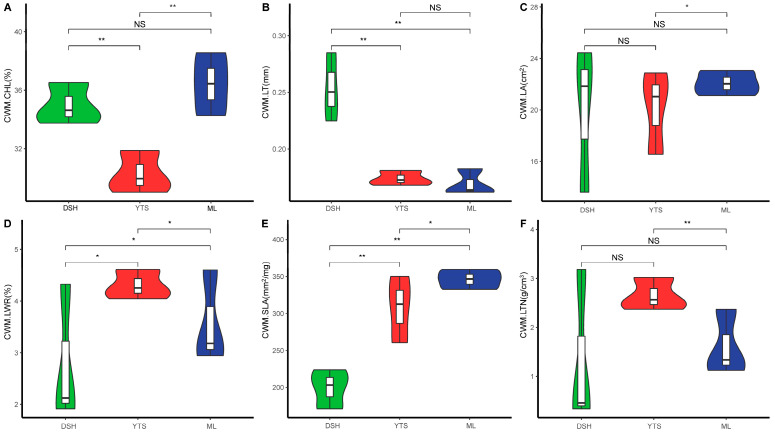
Plot of variation in leaf functional traits in the climax community at different latitudes. In the statistical analysis, a one-way ANOVA was utilized for comparative assessments. The symbol * indicates significant differences (*p* < 0.05), ** indicates highly significant differences (*p* < 0.01), and NS denotes non-significant differences. Three individuals were measured per replicate, with three replicates for each category and three replicates for each plant community. CWM.CHL (**A**)—community-weighted chlorophyll content, CWM.LT (**B**)—community-weighted leaf thickness, CWM.LA (**C**)—community-weighted leaf area content, CWM.LWR (**D**)—community-weighted leaf aspect ratio. CWM.SLA (**E**)—cluster-weighted specific leaf area content, CWM.LTN (**F**)—cluster-weighted leaf tissue density.

**Figure 3 plants-14-00183-f003:**
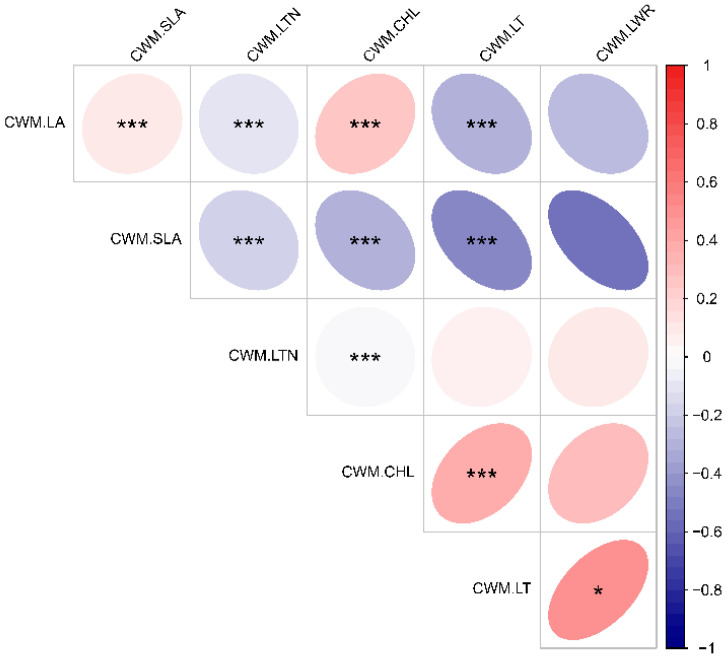
Correlation plots of leaf functional traits in climax communities at different latitudes. * Indicates significant difference (*p* < 0.05), *** indicates highly significant difference (*p* < 0.001). CWM.CHL—community-weighted chlorophyll content, CWM.LT—community-weighted leaf thickness, CWM.LA—community-weighted leaf area content, CWM.LWR—community-weighted leaf aspect ratio, CWM. SLA-cluster-weighted specific leaf area content, CWM.LTN—cluster-weighted leaf tissue density.

**Figure 4 plants-14-00183-f004:**
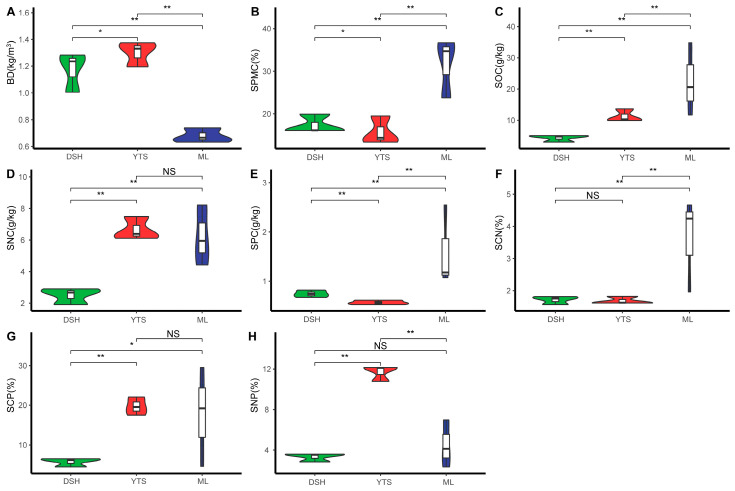
Variation in soil factors in the climax communities at different latitudes. In the statistical analysis, a one-way ANOVA was utilized for comparative assessments. The symbol * indicates significant differences (*p* < 0.05), ** indicates highly significant differences (*p* < 0.01), and NS denotes non-significant differences. Three individuals were measured per replicate, with three replicates for each category. BD (**A**)—soil bulk density, SPMC (**B**)—soil water content, SOC (**C**)—soil organic carbon content, SNC (**D**)—soil total nitrogen content, SPC (**E**)—soil total phosphorus content, SCN (**F**)—soil carbon to nitrogen ratio, SCP (**G**)—soil carbon to phosphorus ratio, SNP (**H**)—soil nitrogen to phosphorus ratio. Soil nitrogen–phosphorus ratio.

**Figure 5 plants-14-00183-f005:**
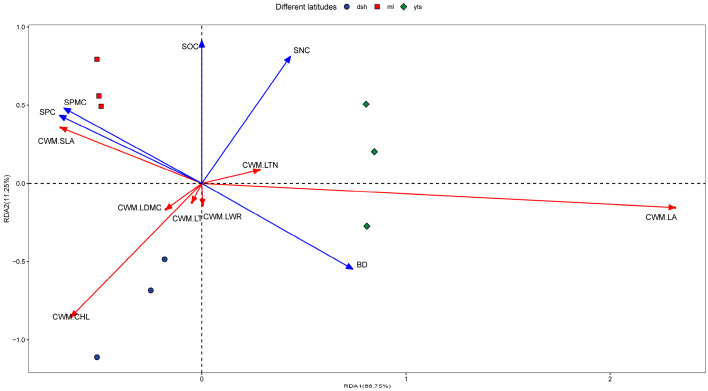
RDA ordination of plant community functional traits with soil factors. The red line is the plant functional trait and the blue line is the environmental factor.

**Figure 6 plants-14-00183-f006:**
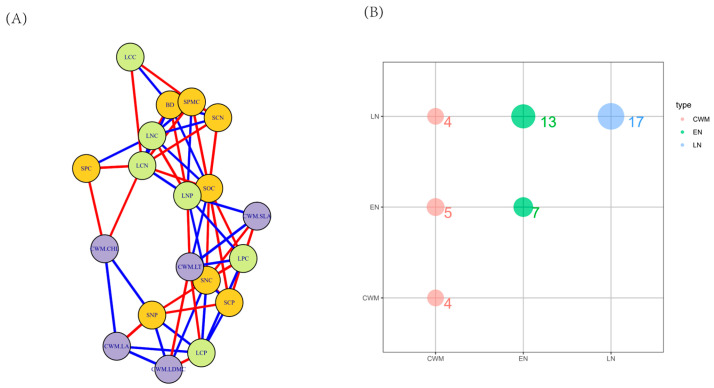
Network relationships of community functional traits with leaf nutrients and soil factors. LCC—leaf organic carbon content, LNC—leaf total nitrogen content, LPC—leaf total phosphorus content, LCN—leaf C:N, LCP—leaf C:P, LNP—leaf N:P, meanings of the rest of the letters refer to Figure 2 and Figure 4. In (**A**), the red line indicates a positive correlation, and the blue line indicates a negative correlation. cwm stands for community functional traits, LN—leaf nutrient content, EN—soil factor. In (**B**), the numbers surrounding the bubbles indicate the total count of correlation links between the variables.

**Table 1 plants-14-00183-t001:** Basic situation of plots.

Sample Site	Dominant Species	Cover (%)	Elevation (m)	Longitude (°)	Latitude (°)
DSH	*Fagus longipetiolata*, *Cunninghamia lanceolata*, *Woonyoungia septentrionalis*, *Synedrella nodiflora*	85	645.58	107.576389 E	29.102778 N
YTS	*Liquidambar formosana*, *Rhus chinensis*, *Styrax confusus*, *Pteridium aquilinum*	80	667.83	108.116208 E	27.110187 N
ML	*Cornus wilsoniana*, *Lindera communis*, *Nandina domestica*, *Selaginella tamariscina*	90	744.35	108.022222 E	25.258333 N

**Table 2 plants-14-00183-t002:** Functional trait indicator selection.

Functional Feature Type	Data Type	Attribute
Leaf thickness (LT)	numerical value	Leaf blade thickness (mm)
Leaf area (LA)	numerical value	Mean leaf blade area of species (cm^2^)
Specific leaf area (SLA)	numerical value	Fresh leaf area/leaf dry mass (mm^2^/mg)
Chlorophyll (CHL)	numerical value	Leaf blade chlorophyll content (%)
Leaf aspect ratio (LWR)	numerical value	Leaf blade length to width ratio (%)
Leaf tissue density (LTN)	numerical value	Leaf blade dry weight/leaf volume (g/cm^3^)

## Data Availability

The original contributions presented in the study are included in the article; further inquiries can be directed to the corresponding author.

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
