# Peer review of "Response of Plant Leaf Traits to Environmental Factors in Climax Communities at Varying Latitudes in Karst Regions"

_plants, 2025, doi:10.3390/plants14020183_

Round 1
Reviewer 1 Report
Comments and Suggestions for Authors
The authors have conducted interesting and meaningful research in the karst regions of southern China. The study of plant trait gradient analysis on this special geomorphic type has been one of the research hotspots in community ecology. However, before the manuscript was finally accepted for publication, the authors had to make substantial changes to the relevant content of the article in order to clarify some confusing issues.
-The Abstract is too long, at least 1/3 shorter is best. The current abstract shows too much specific data, which makes the abstract seem lengthy and lacking in points.
-The method part. I am not sure how the nine 10*10 small plots are set in the 30*30 sample plots, are they randomly arranged? The puzzle is why there are fewer shrub/herb quadrates than tree quadrates?
Is the nutrient content of leaves measured by mixing the dominant species within the plot? Or is it just the edificator?
-L209 The CWM is calculated using abundance weighting. Given that there are multiple growth types of species, I recommend using importance values for weighting. Moreover, the authors has already mentioned the concept of important value when analyzing leaf nutrients. Please also clarify which species can be defined as dominant species? For example, their importance value greater than? Since the number and size of trees, shrubs, and herbaceous quadrats are inconsistent, the authors are asked to identify how they are scaled up to the community level of the same size.
-L265 I do not understand how section 3.2 is relevant to this study. None of this is included in the author's hypothesis. And these contents have nothing to do with experimental treatment at different latitudes. In my opinion, it only proves that the functional traits studied by the authors are redundant. If so, the trait dimensions chosen by the authors are obviously insufficient. In particular, similar content is also involved in Fig.6 and its related explanations.
-Other issues
L18 Delete "as latitude increased"
Tab1 Please change the "、" sign to "," longitude and latitude please add "E" and "N". Do the four dominant species in the table cover different growth forms?
Tab2 detele "data type"column. Please add full names for functional traits, such as leaf area.
L184 Please list the names and abbreviations of all soil factors. Although the author has given this information in the abstract, note that the text and the abstract are separate parts.
Fig.4 I recommend capitalizing things like "dsh" in the graph so that they are unified with Tab1. You may need to include the latitudes of different study sites in the notes, even if they are described in the table. The tab. and fig. are self-explanatory, and they are independent of each other.
Fig.5 The resolution is too low. The indicators used by the authors for RDA and hierarchical analysis are inconsistent. In RDA, the authors used direct soil nutrient indicators such as SPC. However, in the hierarchical analysis, the authors used SNP, SCP, SCN and other proportional indicators, and the conclusions reached were inconsistent. I don't think that's a good thing to do. In addition, the relationship between soil factors and RDA1 is not clear, although RDA1 is related to leaf area and other traits, and it is very good to distinguish the communities of different study sites, but it is necessary to consider whether this relationship is caused by different species composition in different region.
Author Response
Thank you very much for taking time out of your busy schedule to review this manuscript. We appreciate all of your comments and suggestions. We have to say that the reviewers have read the manuscript in detail and made very valuable comments, and we deeply apologise for some of the low-level errors in the manuscript. Therefore, we have revised the manuscript and responded to your questions in a focused manner, please refer to the revised manuscript. The following are the key questions.
Q1: The Abstract is too long, at least 1/3 shorter is best. The current abstract shows too much specific data, which makes the abstract seem lengthy and lacking in points.
Response:We have shortened the abstract section(L10-L30).
Q2: The method part. I am not sure how the nine 10*10 small plots are set in the 30*30 sample plots, are they randomly arranged? The puzzle is why there are fewer shrub/herb quadrates than tree quadrates?
Response:I apologize for the lack of clarity in our description. One of the sample plots we selected measures 900 square meters, which allows us to divide it into nine smaller plots of 100 square meters each. This division facilitates the survey process. Additionally, the categorization of shrubs and grasslands was conducted using field survey methods, as outlined by Fang et al. You can refer to the schematic diagram below; however, we have chosen not to include this diagram in the manuscript to avoid an excess of images.
Q3: Is the nutrient content of leaves measured by mixing the dominant species within the plot? Or is it just the edificator?
Response:Yes, leaf nutrient content is assessed by analyzing the dominant species within the plot. Due to the impracticality of measuring nutrients for all species in the sample plots, some species were present in such low numbers that they fell below the minimum threshold for nutrient measurements, making it impossible to obtain a complete nutrient profile.
Q4: L209 The CWM is calculated using abundance weighting. Given that there are multiple growth types of species, I recommend using importance values for weighting. Moreover, the authors has already mentioned the concept of important value when analyzing leaf nutrients. Please also clarify which species can be defined as dominant species? For example, their importance value greater than? Since the number and size of trees, shrubs, and herbaceous quadrats are inconsistent, the authors are asked to identify how they are scaled up to the community level of the same size.
Response:The reviewers' comments are greatly appreciated. In fact, we also assigned importance values, which are consistent with our results and indicate the functional traits of the dominant species in the community. However, we did not initially describe the importance values, which caused confusion for the reviewers, and we apologize for that oversight. We have now included this information in the manuscript (L211-L213).
Q5: L265 I do not understand how section 3.2 is relevant to this study. None of this is included in the author's hypothesis. And these contents have nothing to do with experimental treatment at different latitudes. In my opinion, it only proves that the functional traits studied by the authors are redundant. If so, the trait dimensions chosen by the authors are obviously insufficient. In particular, similar content is also involved in Fig.6 and its related explanations.
Response:We have carefully reviewed this section, and it aligns with the reviewers' comments. Consequently, we have removed this section from the manuscript.
Q6: L18 Delete "as latitude increased"
Response:We have revised it in the manuscript(L17).
Q7: Tab1 Please change the "、" sign to "," longitude and latitude please add "E" and "N". Do the four dominant species in the table cover different growth forms?
Response:We have revised it in the manuscript(L144). The four dominant species listed in the table represent various growth forms that occupy different ecological niches.
Q8: Tab2 detele "data type"column. Please add full names for functional traits, such as leaf area.
Response:We have revised it in the manuscript(L202).
Q9: L184 Please list the names and abbreviations of all soil factors. Although the author has given this information in the abstract, note that the text and the abstract are separate parts.
Response:We have revised it in the manuscript(L163-L164,L172-L173).
Q10: Fig.4 I recommend capitalizing things like "dsh" in the graph so that they are unified with Tab1. You may need to include the latitudes of different study sites in the notes, even if they are described in the table. The tab. and fig. are self-explanatory, and they are independent of each other.
Response:We have revised it in the manuscript(Firure2,Figure4).
Q11: Fig.5 The resolution is too low. The indicators used by the authors for RDA and hierarchical analysis are inconsistent. In RDA, the authors used direct soil nutrient indicators such as SPC. However, in the hierarchical analysis, the authors used SNP, SCP, SCN and other proportional indicators, and the conclusions reached were inconsistent. I don't think that's a good thing to do. In addition, the relationship between soil factors and RDA1 is not clear, although RDA1 is related to leaf area and other traits, and it is very good to distinguish the communities of different study sites, but it is necessary to consider whether this relationship is caused by different species composition in different region.
Response:We sincerely apologize for this issue. Unfortunately, we did not identify this problem during the data processing phase. The error was caused by a covariance of environmental factors, which resulted in the following warning appearing in our R language analysis: ‘Some constraints or conditions were aliased because they were redundant. This can happen if terms are linearly dependent (collinear)’. This mistake was overlooked by our team. We have now reanalyzed the data while retaining the non-collinear variables. Consequently, we no longer examine specific variable contributions; instead, we focus more on the environmental effects on plant functional traits (L307-L309,L318-L325).
We would like to express our gratitude to the reviewers for their valuable suggestions on the manuscript, which we find to be highly constructive! Finally, we are very grateful to the reviewer for reading the manuscript and putting forward very meaningful suggestions. This is a great help to our article, and we are deeply grateful for it.

Reviewer 2 Report
Comments and Suggestions for Authors
General Comments
The manuscript is not prepared according to the Plants' Journal Instructions (order of the sections) and Abstract currently contains over 500 words, while it should be 200 words maximum. Figure captions are too long, containing methodology in some cases, and not separated from main text, making the manuscript hard to read.
On several places (starting on Lines: 126, 344, 357, 426) in the manuscript there are part of the text that remained after some language editing (“Reason: The revisions improve clarity, enhance technical accuracy, and correct grammatical and punctuation errors while maintaining the original meaning of the text.”) which is really strange given the number of Authors to whom writing of the manuscript were contributed.
There is a lack of explanation in the Methods which methods were used for some of the analyses (e.g. chlorophyll content; leaf volume; direct measurement of carbon, nitrogen and phosphorus in the leaves). In general, Methods section does not proved sufficient details that could enable other researchers to repeat the analyses. It is unclear from which species (in terms of trees-shrubs-herbaceous) leaves were collected and analysed. Adding supplementary files with detailed list of the species present (and analysed) from my point of view is mandatory, to provide foundations for presented results and enable other researchers to compare their results with here presented ones.
There are inconsistencies in used terminology (e.g. for vegetation: climax; top-stage; apical-stage; vertex) and for used abbreviations (LTD vs LTN for leaf blade dry weight / leaf volume; Table 2. Vs. Fig. 2 and Fig. 3.) throughout the manuscript, which makes it hard to follow.
Specific comments
Abstract – at present, Abstract is excessively long and had to be shortened substantially, hence I will not comment on it.
Keywords – „Different latitudes“ is vague expression. You may consider using „Latitudinal gradient“ or similar, instead.
Line 57 – I am confused why are you using „restoration“ here when writing about the climax communities. Climax community can develop during some restoration process, but in general climax communities develops as a result of long-term succession processes, and are not limited to degraded habitats only.
Line 58 – this goes for other climax communities apart the “forest ecosystems”.
Line 67 – I am confused with this: “the adaptation of habitat heterogeneity and plant traits”. Do you meant “the adaptation of plant traits to habitat heterogeneity”?
Line 76 – Please consider using “surveys” or “researches” instead of “discussions”
Lines 117-118 – You should annual average temperature for Dashahe here, to be consistent eith data presented for other two regions surveyed
Sample setup (Lines 140-148) – It is unclear for me for what purpose exactly this setup was used. Have you recorded all species present in these plots/subplots, identify dominant ones that were used for further analyses. This is connected to unclear part of the following section “Plant leaf sampling and processing (Lines 151-169)” and should be rewritten to be more specific and clear.
Lines 164-165 – What do you mean here with “slices”? You should use whole leaf blade for measuring those traits. It is not acceptable to write in the methods “ and other related metrics”. You should write specifically all methods you have used, and later presented in the results. Same goes for “other elements” in the Line 169. If you have measured something and not presented it in the manuscript, there is no need to mention it. Hence, methods and results should be perfectly synchronised with respect to characteristics/parameters analysed and presented in the manuscript.
Line 171 – what is “flora road”?
Line 178 and 181 – So, you have measured only water content not the water capacity? The former can be influenced significantly with field conditions, whether some of your samples were taken after rainy events or longer dry periods.
Line 178 – again “other analyses”
Lines 213-214 – As far as I know chi-square test is not used for testing normality.
Line 216 – why are you mentioning here “leaf decay”? You have not mentioned it earlier in the manuscript.
Line 220 – similar to previous comment. If I understood well your survey you were dealing with fresh leaves and not with “leaf litter”!
Line 223 – which stages of “recovery”? Recovery of what? I was thinking that you have analysed climax vegetation.
Figure 2. – Caption is too long. What do you mean with “ecostatistical significance”? What do you mean here with “life category”? Functional traits or something else? Lettering could be added in part where you describe meaning of the abbreviations, there is no need for extra sentence at the end of the caption. Explanation (or legend) for boxplots within violing graphs is missing.
Lines 253-254 – “The distribution of chlorophyll was more dispersed…” sounds vague. Are you referring here to the difference of measured values between different regions, or to distribution of measured values within region (which will be influenced by the normal or non-normal distribution of it)?
Lines 281-295 – This section is referring to the Fig. 4, but nowhere in the text it is indicated.
Figure 4. – similar comments as for the Figure 2.
Figure 5. – RDA graph should be standalone figure, because at present it is almost impossible to read its content. To me it is unclear from where you obtained values presented in the Fig. 5 (B). Furthermore, Only five soil factors are visible in the ordination graph, while in the 5(B) you presented values for all seven! SNP which has highest value is not visible in RDA graph!?!
Figure 6. – Liine 322 – where are those “red numbers in parentheses”?
Lines 325-327 – You have not shown the overall explained variance obtained with the RDA analyses. Here mentioned 77.27% is one of the first axe.
Lines 335-338 – I cannot see in the presented RDA graph foundation for these claims. What is “community area”?
Lines 362-365 – This part of the text is misplaced here. It sounds like part of the Introduction section. It is not appropriate to start your discussion with it.
Discussion – Given the fact that in my opinion significant part of the Methods and Results has to be revised, I do not see a rationale to go into details in Discussion part of the manuscript. Just to mention two details I have noticed:
Lines 446-447 – “high and low SLA” are not characteristics of the “conservative strategy”. Only low SLA is!
Line 464 – It is unclear why are you mentioning here that mid-latitude regions are more susceptible to human activities. You have not dealt with human activities in your manuscript or mentioned it earlier.
Author Response
Thank you very much for taking time out of your busy schedule to review this manuscript. We appreciate all of your comments and suggestions. We have to say that the reviewers have read the manuscript in detail and made very valuable comments, and we deeply apologise for some of the low-level errors in the manuscript. Therefore, we have revised the manuscript and responded to your questions in a focused manner, please refer to the revised manuscript. The following are the key questions.
General Comments:
Q1:The manuscript is not prepared according to the Plants' Journal Instructions (order of the sections) and Abstract currently contains over 500 words, while it should be 200 words maximum. Figure captions are too long, containing methodology in some cases, and not separated from main text, making the manuscript hard to read (L10-L30).
Response:We have shortened the summary section and have carefully revised the titles of the charts and graphs(L10-30).
Q2: On several places (starting on Lines: 126, 344, 357, 426) in the manuscript there are part of the text that remained after some language editing (“Reason: The revisions improve clarity, enhance technical accuracy, and correct grammatical and punctuation errors while maintaining the original meaning of the text.”) which is really strange given the number of Authors to whom writing of the manuscript were contributed.
Response:We are very sorry for such a cheap mistake and I have removed it from the manuscript.
Q3: There is a lack of explanation in the Methods which methods were used for some of the analyses (e.g. chlorophyll content; leaf volume; direct measurement of carbon, nitrogen and phosphorus in the leaves). In general, Methods section does not proved sufficient details that could enable other researchers to repeat the analyses. It is unclear from which species (in terms of trees-shrubs-herbaceous) leaves were collected and analysed. Adding supplementary files with detailed list of the species present (and analysed) from my point of view is mandatory, to provide foundations for presented results and enable other researchers to compare their results with here presented ones.
Response:We are very grateful to the reviewers for having this query. In fact, we have already explained the determination of C, N, and P in detail in the section on elemental determination (L156-L164). In addition, we have added the method for the determination of leaf phenotypic traits where the reviewer was confused (L192-L200).
Q4: There are inconsistencies in used terminology (e.g. for vegetation: climax; top-stage; apical-stage; vertex) and for used abbreviations (LTD vs LTN for leaf blade dry weight / leaf volume; Table 2. Vs. Fig. 2 and Fig. 3.) throughout the manuscript, which makes it hard to follow.
Response:We use the term ‘climax’ consistently (L197). Table 2 should read ‘LTN’(L203).
Specific comments:
Q1: Abstract – at present, Abstract is excessively long and had to be shortened substantially, hence I will not comment on it.
Response:We have shortened the summary section(L10-L30).
Q2: Keywords – „Different latitudes“ is vague expression. You may consider using “Latitudinal gradient”or similar, instead.
Response:We accepted the reviewers' comments and revised them in the manuscript(L31).
Q3: Line 57 – I am confused why are you using “restoration” here when writing about the climax communities. Climax community can develop during some restoration process, but in general climax communities develops as a result of long-term succession processes, and are not limited to degraded habitats only.
Response:This is a very good question. In fact, we are also very confused, not knowing whether to choose ‘succession’ or ‘restoration’. As a matter of fact, ‘succession’ represents the state of a relatively primitive plant community, and it expresses the process of ‘from nothing to something’. ‘Restoration’ is the process of re-establishing vegetation after habitat destruction, and it expresses the process of “from something to nothing to something again”. In the case of our study area, it is a process of re-establishment of vegetation after habitat destruction, so we believe that the term ‘restoration’ is more appropriate.
Q4: Line 58 – this goes for other climax communities apart the “forest ecosystems”
Response:We are trying to convey that the parietal community has a significant impact on the maintenance and stability of forest ecosystems.
Q5: Line 67 – I am confused with this: “the adaptation of habitat heterogeneity and plant traits”. Do you meant “the adaptation of plant traits to habitat heterogeneity”?
Response:I'm sorry, this was an error in our expression and we accept the reviewer's comments(L58).
Q6: Line 76 – Please consider using “surveys” or “researches” instead of “discussions”
Response:Many thanks to the reviewers for their comments, which we have revised in the manuscript(L67).
Q7: Lines 117-118 – You should annual average temperature for Dashahe here, to be consistent eith data presented for other two regions surveyed
Response:We have revised it in the manuscript(L113).
Q8:Sample setup (Lines 140-148) – It is unclear for me for what purpose exactly this setup was used. Have you recorded all species present in these plots/subplots, identify dominant ones that were used for further analyses. This is connected to unclear part of the following section “Plant leaf sampling and processing (Lines 151-169)” and should be rewritten to be more specific and clear.
Response:The Sample setup section describes the sample setup and survey methodology for the climax community, and this part of the use is necessary. Our sample survey is not just trees, but also includes shrubs and herbs in the community. We try to write this section as specific and clear as possible(L140-L143).
Q9: Lines 164-165 – What do you mean here with “slices”? You should use whole leaf blade for measuring those traits. It is not acceptable to write in the methods “ and other related metrics”. You should write specifically all methods you have used, and later presented in the results. Same goes for “other elements” in the Line 169. If you have measured something and not presented it in the manuscript, there is no need to mention it. Hence, methods and results should be perfectly synchronised with respect to characteristics/parameters analysed and presented in the manuscript.
Response:We apologise for some deviation in our expression. We have made corrections in the manuscript(L156-L164).
Q10: Line 171 – what is “flora road”?
Response:We have revised it in the manuscript(L166-L167).
Q11: Line 178 and 181 – So, you have measured only water content not the water capacity? The former can be influenced significantly with field conditions, whether some of your samples were taken after rainy events or longer dry periods.
Response:This is indeed a highly specialized question. Soil moisture content is significantly influenced by site conditions; however, we needed to measure this indicator to reflect the soil moisture content of the study area at a specific time. For this reason, our sampling period was concentrated in May and June, when rainfall is lower and soil moisture content is less affected.
Q12: Line 178 – again “other analyses”
Response:We have revised it in the manuscript(L172).
Q13: Lines 213-214 – As far as I know chi-square test is not used for testing normality.
Response:We have revised it in the manuscript(L215).
Q14: Line 216 – why are you mentioning here “leaf decay”? You have not mentioned it earlier in the manuscript
Response:We have revised it in the manuscript(L217-L218)
Q15: Line 220 – similar to previous comment. If I understood well your survey you were dealing with fresh leaves and not with “leaf litter”!
Response:We sincerely apologize for the misdescription; we have revised it in the manuscript(L220-221).
Q16: Line 223 – which stages of “recovery”? Recovery of what? I was thinking that you have analysed climax vegetation.
Response:We sincerely apologize for the misdescription; we have revised it in the manuscript(L220-L227).
Q17: Figure 2. – Caption is too long. What do you mean with “ecostatistical significance”? What do you mean here with “life category”? Functional traits or something else? Lettering could be added in part where you describe meaning of the abbreviations, there is no need for extra sentence at the end of the caption. Explanation (or legend) for boxplots within violing graphs is missing.
Response:Our description of this section lacks clarity; therefore, we will revise it in detail in the manuscript(L233-L241).
Q18: Lines 253-254 – “The distribution of chlorophyll was more dispersed…” sounds vague. Are you referring here to the difference of measured values between different regions, or to distribution of measured values within region (which will be influenced by the normal or non-normal distribution of it)?
Response:Our description should be“Chlorophyll distribution was more widely dispersed in the same regional assay”(L254-L255).
Q19: Lines 281-295 – This section is referring to the Fig. 4, but nowhere in the text it is indicated.
Response:We have revised it in the manuscript(L279).
Q20: Figure 4. – similar comments as for the Figure 2.
Response:We have revised it in the manuscript(L295-L303).
Q21: Figure 5. – RDA graph should be standalone figure, because at present it is almost impossible to read its content. To me it is unclear from where you obtained values presented in the Fig. 5 (B). Furthermore, Only five soil factors are visible in the ordination graph, while in the 5(B) you presented values for all seven! SNP which has highest value is not visible in RDA graph!?!
Response:We sincerely apologize for this issue. Unfortunately, we did not identify this problem during the data processing phase. The error was caused by a covariance of environmental factors, which resulted in the following warning appearing in our R language analysis: ‘Some constraints or conditions were aliased because they were redundant. This can happen if terms are linearly dependent (collinear)’. This mistake was overlooked by our team. We have now reanalyzed the data while retaining the non-collinear variables. Consequently, we no longer examine specific variable contributions; instead, we focus more on the environmental effects on plant functional traits (L307-L309,L318-L325).
Q22: Figure 6. – Liine 322 – where are those “red numbers in parentheses”?
Response:We have revised it in the manuscript(L316-L317).
Q23: Lines 325-327 – You have not shown the overall explained variance obtained with the RDA analyses. Here mentioned 77.27% is one of the first axe.
Response:We have revised it in the manuscript(L320).
Q24: Lines 335-338 – I cannot see in the presented RDA graph foundation for these claims. What is “community area”?
Response:We have revised it in the manuscript(L328-L333).
Q25: Lines 362-365 – This part of the text is misplaced here. It sounds like part of the Introduction section. It is not appropriate to start your discussion with it.
Response:We accept the reviewer's comment to add this section to the Introduction section(L36-L42).
Q26: Lines 446-447 – “high and low SLA” are not characteristics of the “conservative strategy”. Only low SLA is!
Response:We have revised it in the manuscript(L435).
Q27: Line 464 – It is unclear why are you mentioning here that mid-latitude regions are more susceptible to human activities. You have not dealt with human activities in your manuscript or mentioned it earlier.
Response:Here, we must clarify that the sample plots chosen in the mid-latitudes are situated near areas of human activity, exhibiting signs of grazing, logging, and other disturbances. We did not address this in the sample site selection section, as it is a subjective factor that may influence the results.
We would like to express our gratitude to the reviewers for their valuable suggestions on the manuscript, which we find to be highly constructive! Finally, we are very grateful to the reviewer for reading the manuscript and putting forward very meaningful suggestions. This is a great help to our article, and we are deeply grateful for it.

Reviewer 3 Report
Comments and Suggestions for Authors
This novel study provides information about the adaptation of plant species in a specific region in response to different environments. However, it needs minor changes to make it acceptable and publish it as a research article.
Comments:
Abstract is too long; summarise it.
Line 65: Describe Karst area complexity and fragility in detail
Line 106: What is meant by top-stage plants? Are these plants in abundance in the Karst region?
Table 1: Add common names along with botanical names of plant species
If possible, provide data related to each specie abundance in a particular latitude
Table 2: Provide the name of the functional traits in addition to their abbreviation, i.e., specific leaf area (SLA).
Line 383-384: This statement is confusing. I don’t think sunlight availability is reduced in lower altitudes that causes a decrease in leaf thickness. Further, plants under high altitude, as you mentioned in the above statement (375–377), are under stress, so technically, high-altitude plants are under stress, but results showed high leaf thickness.
412-414: Why are SCN, SCP, and SNP of karst soils lower than global forest soils or Chinese soils? Is it related to plantation diversity or linked to climactic conditions?
The discussion portion needs improvement as it lacks scientific/logical reasoning to explain results.
Author Response
Thank you very much for taking the time out of your busy schedule to review this manuscript. We appreciate all your comments and suggestions. We have revised the manuscript and response your questions in highlight, please see the revised manuscript. Here were the main questions.
Q1: Abstract is too long; summarise it.
Response: We have shortened the summary section and have carefully revised the titles of the charts and graphs(L10-30).
Q2: Line 65: Describe Karst area complexity and fragility in detail.
Response: We have described it in detail in the manuscripts(L36-L42).
Q3: Line 106: What is meant by top-stage plants? Are these plants in abundance in the Karst region?.
Response: This section is where we describe the error. We have revised it in the manuscript(L97)
Q4: Table 1: Add common names along with botanical names of plant species
Response: We have described it in detail in the manuscripts(Table 1).
Q5: If possible, provide data related to each specie abundance in a particular latitude.
Response: In fact, we can calculate species abundance. However, we primarily analyzed plant functional traits using community weighting, which was derived from the calculation of species importance values (L211-L213).
Q6: Table 2: Provide the name of the functional traits in addition to their abbreviation, i.e., specific leaf area (SLA).
Response: We have described it in detail in the manuscripts(Table 2).
Q7: Line 383-384: This statement is confusing. I don’t think sunlight availability is reduced in lower altitudes that causes a decrease in leaf thickness. Further, plants under high altitude, as you mentioned in the above statement (375–377), are under stress, so technically, high-altitude plants are under stress, but results showed high leaf thickness.
Response: We're very sorry, here's an error in our description that we've corrected in the manuscript. In the context of universal laws, this may appear quite confusing. However, in karst regions, the situation is influenced not only by latitude, which is just one significant factor. In our study, the variation in latitudinal gradients leads to changes in environmental factors, creating habitat heterogeneity within the study area. Consequently, our findings may seem less reasonable; however, the presentation of these results reflects the response of plant community functional traits to environmental factors (L365-L379).
Q8: 412-414: Why are SCN, SCP, and SNP of karst soils lower than global forest soils or Chinese soils? Is it related to plantation diversity or linked to climactic conditions?
Response: We compared our results with global soils and Chinese soils. At small regional scales, plant traits may be more significantly influenced by the local environment, whereas at larger regional scales, climatic factors may play a more dominant role.
Q9: The discussion portion needs improvement as it lacks scientific/logical reasoning to explain results.
Response: In fact, we identified several errors in our analyses within the Results section. We have revised these errors in the manuscript and made corresponding updates in the Discussion. The outcomes of these revisions are highlighted in the manuscript.
Finally, we are very grateful to the reviewers for their numerous constructive comments on the manuscript, which have enhanced the quality and readability of the document. Thank you once again!

Reviewer 4 Report
Comments and Suggestions for Authors
The article is missing one chapter on ecological factors, temperature, precipitation and UV-B radiation, which are key to plant stress and of course affect plant morphology. Because the altitude itself does not affect plant traits, but they are affected by environmental factors that change with altitude (UVB radiation, temperature). If you have measured the data, that would be best, otherwise I suggest data from nearby meteorological stations and a paragraph about how T and UVB radiation change with altitude in principle. It is then necessary to define the aforementioned stress factors in the discussion and add some references. It would also be very good to add photographs of individual locations so that it is comparable with karst around the world. Karst is in different parts of the world, the original word Karst originates from Slovenia. Of course, Karst in Slovenia, Ireland is different from the karst in this article. Therefore, photography is very welcome.

Author Response
Thank you very much for taking the time out of your busy schedule to review this manuscript. We appreciate all your comments and suggestions. We have revised the manuscript and response your questions in highlight, please see the revised manuscript. Here were the main questions.
Q1: The article is missing one chapter on ecological factors, temperature, precipitation and UV-B radiation, which are key to plant stress and of course affect plant morphology. Because the altitude itself does not affect plant traits, but they are affected by environmental factors that change with altitude (UVB radiation, temperature). If you have measured the data, that would be best, otherwise I suggest data from nearby meteorological stations and a paragraph about how T and UVB radiation change with altitude in principle. It is then necessary to define the aforementioned stress factors in the discussion and add some references. It would also be very good to add photographs of individual locations so that it is comparable with karst around the world. Karst is in different parts of the world, the original word Karst originates from Slovenia. Of course, Karst in Slovenia, Ireland is different from the karst in this article. Therefore, photography is very welcome.
Response: We are very grateful to the reviewers for their comments, but we are really sorry that we did not carry out meteorological observations of the study areas, which are in nature reserves, for which the establishment of meteorological stations is not permitted. However, we are grateful for the reviewers' comments and we hope to set up research plots than can be used for meteorological observations in our future research to fully investigate the effects of meteorological factors on plant functional traits at different latitudes and altitude differences. In addition, we undertook a lot of work in the field survey and did not pay special attention to the photography of the sample plots, and we will surely strengthen the photography of the sample plots in our future research.
Finally, we are very grateful to the reviewers for their numerous constructive comments on the manuscript, which have enhanced the quality and readability of the document. Thank you once again!

Reviewer 5 Report
Comments and Suggestions for Authors
This paper provides an insightful exploration of how plant functional traits in karst climax communities vary across latitudinal gradients and how these traits are influenced by environmental factors. By analyzing data from three distinct sites in Guizhou Province, the authors reveal significant patterns in traits such as leaf thickness, specific leaf area, and soil nutrient composition. The study effectively demonstrates how latitude and soil characteristics, particularly nitrogen and carbon content, drive functional trait variations. Overall, the paper contributes valuable knowledge to understanding plant-environment interactions in karst ecosystems, offering implications for ecological restoration and management in these unique habitats.
The manuscript is well-structured, with results clearly presented and supported by a thorough discussion, leading to strong and well-founded conclusions. I recommend accepting the manuscript after minor revisions.
Specific Comments:
Lines 116–117: Please briefly explain the criteria used for selecting the climax communities.
Author Response
Thank you very much for taking the time out of your busy schedule to review this manuscript. We appreciate all your comments and suggestions. We have revised the manuscript and response your questions in highlight, please see the revised manuscript. Here were the main questions.
Q1: Lines 116–117: Please briefly explain the criteria used for selecting the climax communities..
Response: We established two criteria for selecting the apex community: first, our research team had previously studied these plots; second, the biomass of the apex community was the most significant in the entire forest, exhibiting greater species richness, and the plants were larger and less individually differentiated. We have added this section to the manuscript (Ll105-L108)。
Finally, we are very grateful to the reviewers for their numerous constructive comments on the manuscript, which have enhanced the quality and readability of the document. Thank you once again!

Round 2
Reviewer 1 Report
Comments and Suggestions for Authors
In the current version, the authors have made changes or detailed replies to my concerns. Now I don't think there's much wrong with the manuscript. The author needs to make some minor changes to some errors in the text before being considered for acceptance. For example, the "、"in Table 1 has not been replaced with", ". Another example is the format of the reference, such as ref1, the abbreviation of the journal is incorrect; ref47 and ref48, if it is a Chinese journal, it is hoped that the authors will mark "in Chinese with English abstract".
Author Response
Thank you very much for taking time out of your busy schedule to review this manuscript. We appreciate all of your comments and suggestions. We have to say that the reviewers have read the manuscript in detail and made very valuable comments, and we deeply apologise for some of the low-level errors in the manuscript. Therefore, we have revised the manuscript and responded to your questions in a focused manner, please refer to the revised manuscript. The following are the key questions.
Q1: In the current version, the authors have made changes or detailed replies to my concerns. Now I don't think there's much wrong with the manuscript. The author needs to make some minor changes to some errors in the text before being considered for acceptance. For example, the "、"in Table 1 has not been replaced with", ". Another example is the format of the reference, such as ref1, the abbreviation of the journal is incorrect; ref47 and ref48, if it is a Chinese journal, it is hoped that the authors will mark "in Chinese with English abstract".
Response:We are grateful to the reviewers for their insightful comments. We have revised Table 1 and updated the references accordingly. Additionally, we have added ‘In Chinese with English abstract’ to all references pertaining to Chinese Journals. (L144,L505-L600)
We would like to express our gratitude to the reviewers for their valuable suggestions on the manuscript, which we find to be highly constructive! Finally, we are very grateful to the reviewer for reading the manuscript and putting forward very meaningful suggestions. This is a great help to our article, and we are deeply grateful for it.
